# Surveillance for Enteroviruses Associated with Hand, Foot, and Mouth Disease, and Other Mucocutaneous Symptoms in Spain, 2006–2020

**DOI:** 10.3390/v13050781

**Published:** 2021-04-28

**Authors:** Nieves Martínez-López, Carmen Muñoz-Almagro, Cristian Launes, Ana Navascués, Manuel Imaz-Pérez, Jordi Reina, María Pilar Romero, Cristina Calvo, Montserrat Ruiz-García, Gregoria Megias, Juan Valencia-Ramos, Almudena Otero, María Cabrerizo

**Affiliations:** 1Enterovirus Unit, National Centre for Microbiology, Instituto de Salud Carlos III, 28220 Madrid, Spain; niemarti@ucm.es (N.M.-L.); aotero@isciii.es (A.O.); 2Microbiological and Paediatric Departments, Hospital San Joan de Déu, 08950 Barcelona, Spain; cma@sjdhospitalbarcelona.org (C.M.-A.); claunes@sjdhospitalbarcelona.org (C.L.); 3Microbiological Department, Complejo Hospitalario de Navarra, 31008 Navarra, Spain; ana.navascues.ortega@navarra.es; 4Microbiological Department, Hospital de Basurto, 48013 Bilbao, Spain; MANUEL.IMAZPEREZ@osakidetza.eus; 5Microbiological Department, Hospital Son Espases, 07020 Palma de Mallorca, Spain; jorge.reina@ssib.es; 6Microbiological and Paediatric Departments, Hospital La Paz, 28220 Madrid, Spain; mpromero.hulp@salud.madrid.org (M.P.R.); ccalvorey@gmail.com (C.C.); 7Microbiological Department, Hospital de Elche, 03203 Alicante, Spain; ruiz_mongar@gva.es; 8Microbiological and Paediatrics Department, Complejo Hospitalario de Burgos, 09006 Burgos, Spain; gmegias@saludcastillayleon.es (G.M.); jvalenra@gmail.com (J.V.-R.)

**Keywords:** exanthema, genotyping, coxsackievirus A6, enterovirus A71, phylogenetic analysis

## Abstract

Hand, foot, and mouth disease (HFMD) is a mild illness caused by enteroviruses (EV), although in some Asian countries, large outbreaks have been reported in the last 25 years, with a considerable incidence of neurological complications. This study describes epidemiological and clinical characteristics of EV infections involved in HFMD and other mucocutaneous symptoms from 2006 to 2020 in Spain. EV-positive samples from 368 patients were included. EV species A were identified in 85.1% of those typed EV. Coxsackievirus (CV) A6 was the prevalent serotype (60.9%), followed by EV-A71 (9.9%) and CVA16 (7.7%). Infections affected children (1–6 years old) mainly, and show seasonality with peaks in spring–summer and autumn. Clinical data indicated few cases of atypical HFMD as well as those with neurological complications (associated with the 2016 EV-A71 outbreak). Phylogenetic analysis of CVA6 VP1 sequences showed different sub-clusters circulating from 2010 to present. In conclusion, HFMD or exanthemas case reporting has increased in Spain in recent years, probably associated with an increase in circulation of CVA6, although they did not seem to show greater severity. However, EV surveillance in mucocutaneous manifestations should be improved to identify the emergence of new types or variants causing outbreaks and more severe pathologies.

## 1. Introduction

Hand, foot, and mouth disease (HFMD) is a limited acute infection produced by enteroviruses (EV), principally from species A (EV-A). Transmission is faecal–oral or through respiratory secretions and contact with cutaneous lesions, and infections are prevalent in children younger than 5 years old [1,2,3]. Typical clinical manifestations are vesicular lesions in the palm of the hands, feet, and in the oral mucosa. HFMD can lead to fever, but subsequent neurological complications are infrequent, except for the large outbreaks occurring every 2–3 years in countries from the Asia-Pacific region since 1997, where there is a considerable incidence of neurological and cardiopulmonary cases [4,5,6]. Other frequent mucocutaneous manifestations caused by EV are herpangina and non-specific exanthemas [1,6].

EV-A71 and coxsackievirus (CV) A16 were the most common serotypes causing HFMD, although other EV-As can also be implicated [3,4]. EV-A71 is the main EV type associated with neurological complications after HFMD in the Asia-Pacific region, which is why a HFMD surveillance system has been established for 20 years and it is considered to be a mandatory reporting disease in many countries from that region [5,6,7,8]. In the last few years, two other CVA types have emerged as responsible for these pathologies, CVA10 and CVA6 [9,10]. Since 2008, many outbreaks were reported due to these viruses in different parts of the world. CVA6, also being causative of the atypical HFMD, manifesting high fever, vesicular lesions on forearms, calf and perioral regions, is more visible than with typical HFMD [11,12,13]. Moreover, between 2009 and 2010 in Japan and Taiwan, a clinical evolution emerged from herpangina to atypical HFMD with perioral eruptions [14,15].

Up until a few years ago, there were no data about the EV serotypes circulating in Europe and causing HFMD, because there is no obligatory notification due to the benign course of infections, as most of them do not require hospitalization. However, the increasing circulation of the emerging types, such as CVA6, firstly detected in an outbreak in Finland during 2008 [16], along with the detection of more severe cases worldwide, and the appearance of accompanying symptoms, such as onychomadesis [15,16,17,18,19,20], suggests the necessity of a surveillance system of HFMD and its clinical evolution in Europe as well [21,22,23].

In Spain, CVA6 started to circulate between 2008 and 2010, and some works have reported outbreaks and disease cases in different regions, associated to typical, non-typical, or onychomadesis presentations [24,25,26,27,28]. However, and with the exception of an interesting study of all cases diagnosed in 5 primary care centres from Barcelona during one year [27], there are no long studies nationwide about the detection and characterization of EV serotypes involved in mucocutaneous diseases. In this retrospective study, the epidemiology of EV detected in these types of pathologies in Spain during a period of 15 years was described, as well as the associated clinical characteristics.

## 2. Materials and Methods

### 2.1. Patients and Clinical Samples

Since 1988, the National Centre for Microbiology (CNM) has been designated as the National Polio Laboratory for surveillance of poliovirus and non-polio enterovirus (NPEV) infections. For NPEV surveillance, Spanish hospitals can send EV-positive samples for type characterization, voluntarily and at the discretion of clinicians. The CNM receives between 500 and 900 specimens each year from patients with EV infection and different clinical manifestations, mainly neurological, cutaneous, or respiratory.

This study included 386 EV-positive samples collected from 368 patients with different mucocutaneous symptoms who attended different Spanish hospitals between January 2006 and March 2020. Typical HFMD was defined by blisters on the palms, soles, buttocks, knees, or elbows [29]; atypical HFMD was diagnosed when the patients presented with a vesiculobullous and erosive eruption involving more than 10% of the body, including the perioral, extremity, or truncal areas; herpangina was characterised by vesicular eruptions only in the mouth or throat; non-specific exanthema was a widespread rash or area of irritated or swollen skin; petechiae were defined by tiny red, flat spots on skin; and, finally, onychomadesis was loosening and shedding of the nails.

EV detection in clinical samples was performed in each hospital using commercial RT-PCR. In 18 cases, two different samples from the same patient were received. The main sample types were vesicular or cutaneous swabs (N = 148, 38.3%), followed by pharyngeal or nasopharyngeal exudates (N = 135, 34.9%), stool (N = 59, 15.2%), and serum samples (N = 44, 11.4%).

### 2.2. EV Genotyping and Phylogenetic Analyses

For EV type characterization, amplification of 3′-part of the VP1 gene was performed using conventional RT-nested PCR, specific for EV-A, EV-B, EV-C, and EV-D68, as previously reported [30,31], followed by sequencing and analysis of the sequences obtained (BLAST, http://blast.ncbi.nlm.nih.gov/, accessed on 27 April 2021). A specific type was assigned when the homology in the nucleotide sequence was higher than 75%.

A phylogenetic analysis was carried-out in the 3′-VP1 region (400 bp) with 147 CVA6 Spanish sequences, the prototype strain, and other sequences from other countries available in GenBank. Alignments were performed with ClustalW [32] and the phylogenetic trees were built with the neighbour-joining method, distance model with the method maximum likelihood, and 1000 pseudo-repeats, such as the boot-strap method, in MEGA program version 10.1.7 (http://www.megasoftware.net, accessed on 8 April 2021).

### 2.3. Statistical Analysis

Quantitative variables were expressed as mean and standard deviations (SD) and the qualitative variables as proportions. For the comparison of proportions, the Chi-square test was used with a 95% confidence interval (http://openepi.com/, accessed on 8 April 2021). *p*-values < 0.05 were considered significant.

## 3. Results

### 3.1. EV Type Characterization

The number of EV-positive samples received each year ranged from one in 2009 to 65 in 2016. Of the total, 88.1% (340/386) were successfully genotyped, revealing the presence of 24 different types (Figure 1). In all cases with two samples, the same serotype was detected in both specimens. Taking into account infections in which EV type was identified (N = 323), the most frequent EV detected was CVA6 accounting for 60.9% of the cases (N = 197), followed by EV-A71 (N = 32, 9.9%), CVA16 (N = 25, 7.7%), and CVA10 (N = 9, 2.8%). Other EV serotypes were identified in minor proportion (Figure 1). Most of the detected EV belonged to species A (275/323, 85.1%), while 47 were EV-B (14.5%) and only in one was EV-C, a CVA11 (0.3%). No EV-D68 was detected.

Regarding annual distribution, EV infections showed seasonality with two peaks, one in spring–early summer (from March to July) with 49.2% of total infections, and another in autumn, October, and November (27.8%) (Figure 2). Detection of the different EV-A types associated with HFMD/exanthemas varies annually as it can see in Figure 1. CVA6 was firstly detected in 2010 and its presence increases every year, accumulating between 33 to 95.8% of annual detections between that year and 2019. Secondly, EV-A71 associated with mucocutaneous pathologies was prevalent in 2016 (37.5%) although it was also detected in 2012, 2015, and 2018. CVA16 was the most frequent type (50% of detections) until 2010 when it was replaced by CVA6. Finally, CVA10 has been detected since 2014, but in very few cases.

The types of samples taken were compared with the results of EV specie detection. In vesicular or cutaneous swabs, 124 out of 275 EV-As (45.1%) and only 6 out of 47 EV-B (12.8%) were detected (*p* < 0.0001). However, EV-Bs were identified more frequently in respiratory or faecal samples than EV-A (31/46, 67.3% vs. 133/276, 48.2%), as well as in serum samples (9/45, 20.0% vs. 19/277, 6.9%) (*p* < 0.05).

### 3.2. Clinical and Epidemiological Characteristics of EV Infections

Most of the 368 patients included in this study were children (86.2%). The male/female proportion was 1.55, which was statistically significant (*p* < 0.0001). Clinical features of the patients and the referred diagnosis provided by the clinicians are indicated in Table 1.

EV-A infections were more frequent in children between 4 days and 6 years-old (236/275, 85.8%), with a mean age of 1.73 ± 1.13SD years old, but they also occurred in adult patients (35/277, 12.6%), with a mean age of 33.41 ± 9.29SD years old. EV-B infections were also prevalent in paediatric patients (45/47, 95.7%), with a mean age of 1.37 ± 1.1SD years old. Furthermore, 86.8% of the total typed infections in patients >15 years (33/38) were caused by CVA6 (*p* < 0.0001). The only EV-C infection detected (CVA11) was in an adult patient.

Comparing the clinical symptoms with the EV type, CVA6 caused the majority HFMD cases (137/215, 63.7%) and was responsible for all but one case of atypical HFMD (13/14, 92.8%) (*p* < 0.0001) (Table 2). However, it was not detected in herpangina cases. In exanthema or rash, CVA6 was identified frequently (37%) as were EV-B types (28.3%). CVA16 and CVA10 caused typical HFMD, herpangina and exanthemas in similar proportions, while EV-A71 was identified practically only in typical HFMD (28/32 EV-A71-positive cases, 87.5%, *p* < 0.0001). Regarding clinical complications, onychomadesis was detected in 16 cases (14 of them during 2011 and 2012), but seven different EV-A types were involved. In five of the seven cases that referred neurologic symptoms after HFMD, EV-A71 was detected (71.4%, *p* < 0.05) (Table 2). A meningitis case (with exanthema) was caused by E-5, and in another child with paralysis and HFMD symptoms, EV type could be genotyped.

During the study-period, CVA6 was also identified in 54 cases with respiratory symptoms or fever, and in 10 cases with CNS pathologies, meningitis or encephalitis. In 50% of these neurological cases (three children and two adults), CVA6 was detected in CSF sample.

### 3.3. Phylogenetic Analysis of CVA6

To investigate the spatiotemporal relationships among CVA6 strains, the 121 VP1 sequences assigned to the enterovirus serotype CV-A6 were compared to 109 homologous sequences from other European and Asian countries available in GenBank and 26 Spanish ones published previously [26]. Sequences from GenBank used in the phylogenetic analysis are summarised in Appendix A (Appendix A).

The resulted phylogenetic tree showed that all Spanish CVA6 included in this study (from 2010 to 2020) are grouped into the same cluster or lineage, along with most of the sequences detected in other geographic regions during the same years, and separated from the groups containing the older strains. Within this lineage, different sub-clusters were observed (Figure 3). All sequences were deposited into the GenBank database (MW717454–MW717574).

## 4. Discussion

HFMD is a paediatric disease mainly caused by EV that in Asian countries produces large outbreaks with a significant number of cases with serious neurological and cardiorespiratory complications, even fatal [4,5,6,7,8]. In Europe and other parts of the world, the same type of outbreak has not been described so far, but the clinical characteristics of this disease and the epidemiology of the serotypes implicated changed in the last 10 years [11,12,13,20,21,22,23,26]. This is the first large study in Spain describing the epidemiology of EV infections associated with mucocutaneous pathologies during a 15-year period from 2006 to 2020.

In our series, different EV serotypes were identified and most of them (85%) belonged to species A. This was expected as classically, the HFMD was associated with EV-A, principally CVA16 and EV-A71. Recently, however, the predominant serotype has been CVA6, with more than 60% of detections, leaving CVA16 and EV-A71 at rates below 10%. These data are in agreement those described in a previous study from our group [26] where the increased detection of CVA6 was already observed, displacing CVA16 as the main cause of HFMD. As in other neighbouring countries, CVA6 emerged in Spain about 10 years ago and is now considered an endemic serotype, being one of the five most frequent EV that circulate every year [22,23]. CVA10 is another EV-A that began to be detected and associated with HFMD in the same period of time, both in Spain and in other countries [4,9,21,23,24]. Although it is detected every year, it does so at low levels, and it does not seem to have the same ability to produce skin lesions as CVA6. This contrasts with other studies that describe the increase of its prevalence in HFMD cases since 2014. In fact, between 2017 and 2018 in Hangzhou, China [18], CVA10 was involved in HFMD to a greater extent than CVA6. With regard to EV-A71, this work confirms that this serotype is not prevalent in cutaneous infections in Spain, and it is more involved in the development of febrile illnesses in young children and neurological pathologies directly [22,26,33,34], as occurs in other European countries [21,23,35]. In fact, EV-A71 was identified in five of the seven cases with neurological complications (meningitis or encephalitis) after the cutaneous eruption, even if these cases were reported during 2016, specifically in the context of a severe encephalitis outbreak reported that year [33,34]. Furthermore, during that outbreak, the number of EV-A71 neurological infections with HFMD or exanthema was very low (4–18% according to our previous studies [33,34]) compared with data from the outbreaks reported in Asia-Pacific countries, where neurological complications occurred after HFMD in most cases [5,6,7,8]. It has been suggested that differences in EV-A71 epidemiology and pathogenicity among world regions can be due to circulation of different strains, since Asian outbreaks have been connected to different subgenogroups (B3, B4, C1, C2, and C4), while in Europe, only EV-A71 C1 and C2 strains have been detected so far [5,26,35,36]. In fact, the encephalitis outbreak that occurred in Spain in 2016 was associated with the emergence of a new C1 variant that had been detected in other European countries one year before, also associated with severe neurological cases [33,34]. Despite findings of emergence of novel genotypes or subgenotypes and recombinant forms giving rise to serious outbreaks, the correlation between variability and virulence in EV71 remains unclear.

Comparing the type of sample collected with the type of EV identified, EV-A or EV-B, data confirmed that the optimal sample for HFMD/exanthema diagnosis was the vesicular or cutaneous swab, since the use of other types of specimens such as faecal or respiratory samples may lead to the identifying of an EV that may not be responsible for this specific pathology. However, these indirect samples can be used for etiological detection if the cutaneous one is not available [37]. Curiously, in the case of serum samples, EV-B were more frequent that EV-A, although all EV-B infections detected in blood caused exanthema and not HFMD.

The number of EV-positive samples associated with HFMD/exanthema received in the lab increased over the years of the study period, until March 2020 where they dropped sharply due to the COVID-19 pandemic, as has occurred with other transmissible diseases [38]. The explanation could be in increased circulation of CVA6 with a more visible clinic manifestations and also greater interest of clinicians in EV surveillance, as has occurred in other European countries in recent years [22,36]. In fact, 2016 was the year with the highest number of EV-positive samples received, probably because, in that year, the above mentioned EV-associated encephalitis outbreak in Spain that alerted paediatricians and microbiologists to these infections occurred [33,34]. In this study, EV infections causing mucocutaneous manifestations showed a biannual distribution, with one peak in spring–summer and another in autumn. Data coincide with those observed in template countries for EV infections overall, including Spain, although the autumn peak is usually not as pronounced [22,26,30]. They are also similar to those described in a study conducted in China [39], in which cases of HFMD by CVA6 seem to have a characteristic peak in autumn–winter that in Spain can be associated with the beginning of the school year.

In our series, most of the patients were children under 6 years of age, regardless of whether the infection was caused by EV-A or EV-B. However, almost all those infections that occurred in adults were caused by CVA6. There results confirmed the increased ability of this virus to produce HFMD in adults, as described in other recent publications [40,41,42].

Regarding the clinical diagnosis that hospitals referred to when they sent the sample for genotyping, in the vast majority of cases, it was HFMD or exanthema, without reporting specific symptoms. HFMD was caused by CVA6 mainly, but other EVs, including EV-B, were implicated in a minor proportion. Surprisingly, atypical HFMD signs were described in only 14 cases, all of them CVA6 positive. In most published studies about CVA6 infections, including some from Spain [11,12,13,20,21,27,28], the clinical presentations associated were non-typical, defined as a varicelliform rash extended to the limbs and face. It is likely that clinicians did not indicate these differential features in many of the HFMD cases sent and, consequently, in our series, atypical HFMD cases have been underestimated. Patients who presented herpangina were also very scarce, a fact that can also be related to the evolution to a more atypical HFMD symptomatology, as has occurred in Asia-Pacific regions since 2010 [14,15], although in our series, herpangina was associated with other CVA serotypes rather than with CVA6. Onychomadesis, or nail fall, after a HFMD episode, which had already been confirmed in previous studies [15,16], is reported in several cases, and although CVA6 was the type more frequently detected in them, other EVs were also identified. Additionally, most of the cases occurred between 2010 and 2012. We do not have an explanation for this fact, although it could be due to a bias in the clinical information received from patients. With respect to those cases with non-specific rashes or exanthemas, EV-A types were prevalent but serotypes from species B were also detected at a high rate, confirming the implication of EVs, such as E-9 or E-11 in these types of cutaneous pathologies [1,6].

Finally, and as already mentioned, neurological cases associated with HFMD symptoms were restricted to EV-A71 infections and to the year 2016. In Asia-Pacific regions, EV-A71 was the main cause of these complications, although in recent years it was also associated with CVA6 infections [19]. According to Yang et al., [19] the greater severity of CVA6 infection was related to the existence of two mutations in the VP1 region: V174I and T283A. These mutations were not found in Spanish strains from our study, nor has an association between CVA6 and neurological disease after exanthematous symptoms been observed. However, CVA6 was detected in CSF from several neurological cases, which did not present HFMD manifestations, suggesting the neurotropic character of this EV. In a previous study [19], circulating CVA6 strains were classified into six lineages (A, B, C, D, E1, and E2). Based on that classification, all Spanish CVA6 strains belong to E2 linage and displayed close temporal relationships to viruses recovered in other countries in Europe and Asia since 2010 to the present. Within this cluster, Spanish sequences from the same year fell into different sub-clusters, consistent with the hypothesis of multiple introductions of distinct CV-A6 strains. In agreement to other studies [21,43], the analysis of the VP1 gene demonstrated that these variants are genetically distinct from other previously identified CVA6 strains (before 2008), but the underlying reason of the rapid emergence worldwide, associated with changes in the phenotype of the CVA6 infections, are unknown. No association between VP1divergence and the different symptoms presented (HFMD, exanthema, fever) was observed in our study. However, Gaunt et al. [44] described that the emergence of new CVA6 recombination groups in non-structural regions (3Dpol) might be related to its spread in Europe and Asia in recent years, and it could have a role in the pathogenicity of CVA6 infections. Unfortunately, phylogenetic analysis in 3Dpol region was not performed here. Although a high variability or severity of CVA6 infections was not observed in Spain in recent years, the epidemiology of the virus could change in the future, as has already occurred in other parts of the world. Consequently, more surveillance studies are needed, as well as more precise information on the associated symptoms, to increase our knowledge of the epidemiology and clinical characteristics of mucocutaneous infections caused by CVA6 and other EVs. There are plans for several hospital-based EV surveillance studies, including for HFMD, to be conducted through the European Non-Polio Enterovirus Network (ENPEN) https://www.escv.eu/enpen/, accessed on 27 April 2021, which will help us in our mission.

## 5. Conclusions

EV serotypes from species A are the main cause of HFMD and non-specific exanthemas in Spain, affecting children between 1 and 6 years old, as well as some adults. CVA6 emerged in 2008 and its circulation has increased since then; it is now currently considered a prevalent serotype. It has not been possible to demonstrate a significant increase in atypical, or more severe, forms of the disease with the exception of some cases associated with EV-A71 during the outbreak of encephalitis that occurred in 2016. CVA6 strains detected between 2010 and 2020 in Spain were similar to those circulating in Europe and Asia during the same study period, which suggests that this serotype has not shown high temporal or geographic variability since then. Surveillance of EV infections must be maintained and strengthened in order to detect the emergence of new viruses or variants associated with outbreaks, as well as more severe pathologies, to be able to act accordingly.

## Figures and Tables

**Figure 1 viruses-13-00781-f001:**
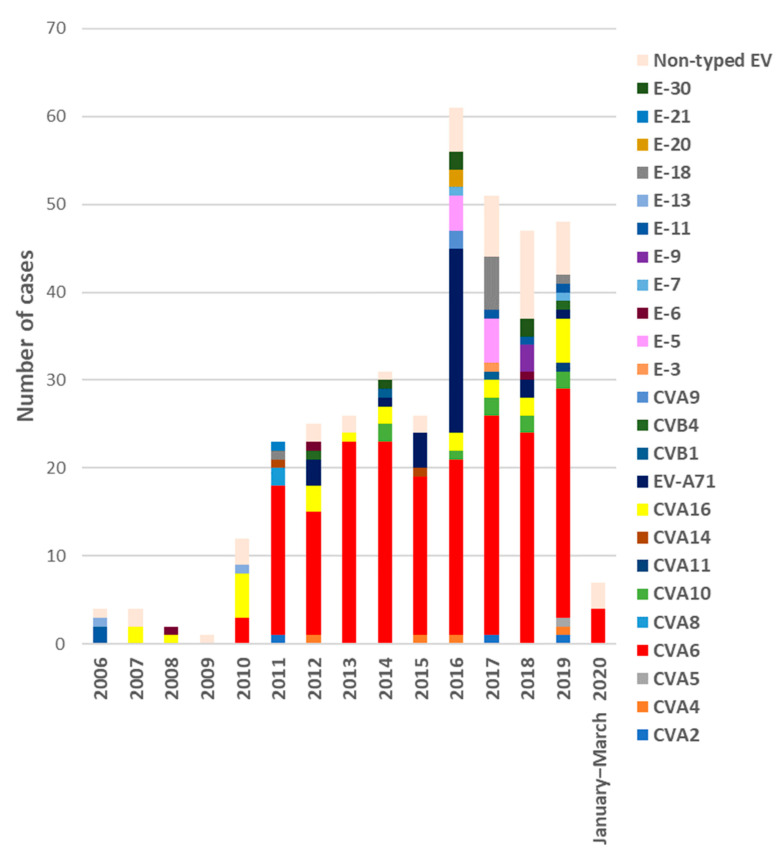
Number of EV-positive samples received from HFMD/exanthemas cases each year during the period of study (January 2006–March 2020) and annual distribution of the different EV types identified.

**Figure 2 viruses-13-00781-f002:**
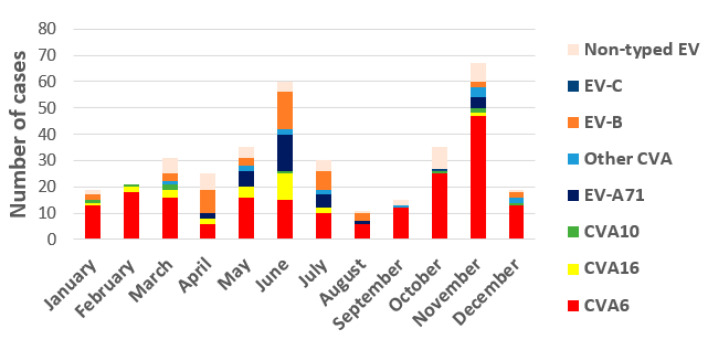
Seasonal circulation of prevalent EV-A, EV-B, EV-C types, and non-typed EV detected in the study.

**Figure 3 viruses-13-00781-f003:**
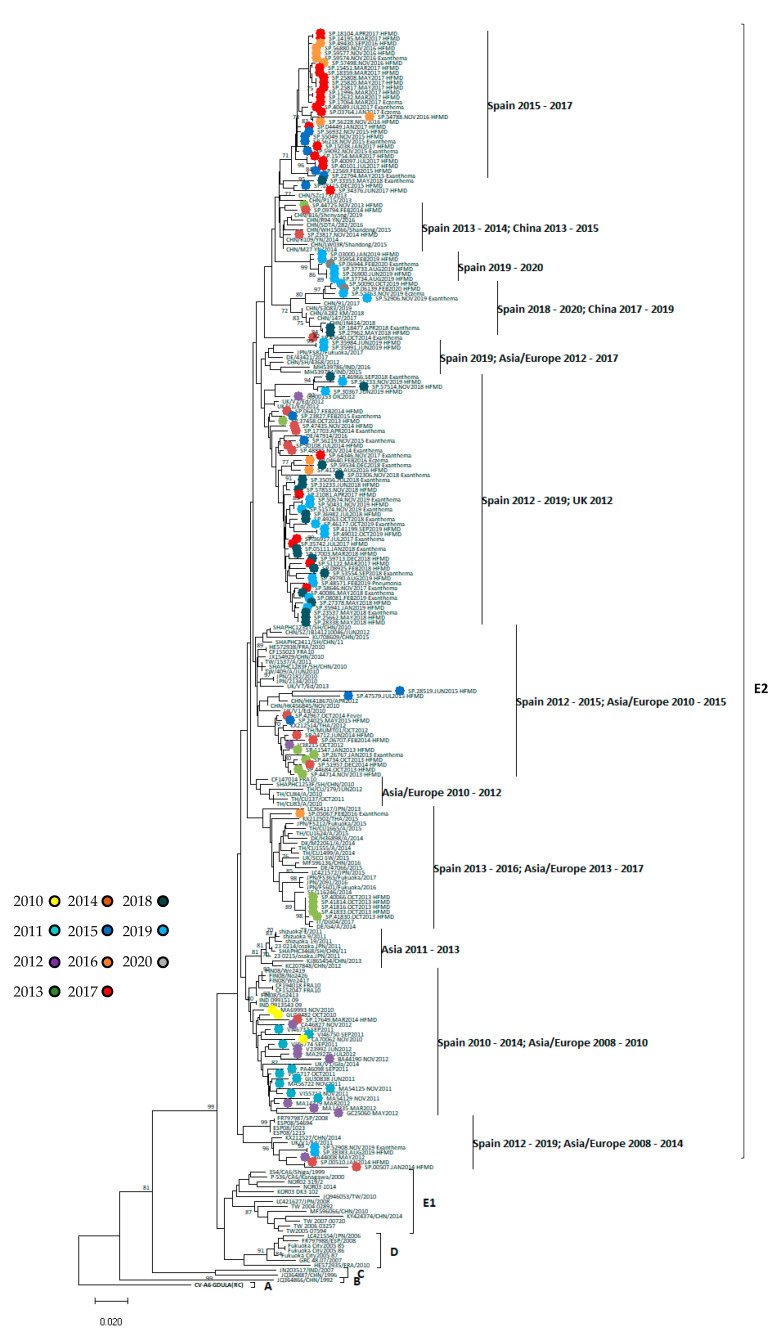
Phylogenetic tree of 3′VP1 region with CVA6 sequences from 1992–2020 classified into six linages, A-E2. Trees were reconstructed using the neighbour-joining method and the maximum composite likelihood model. Tree was rooted with the prototype sequence Gdula (GenBank accession number AY421764). Bootstrap resampling (1000 replicates) was used to determine robustness of groupings. Values of >70% shown. Spanish sequences obtained in this study were coloured according to year of detection (2010–2020).

**Table 1 viruses-13-00781-t001:** Clinical characteristics and symptoms presented by the patients included in the study.

**Epidemiological Features**	
patients < 15 years-old	317 (86.2%)
range	4 days–14 years
mean age	1.87 ± 1.73SD
patients > 15 years-old	51 (13.8%)
range	18-90 years
mean age	36.52 ± 13.82SD
male/female	225/143
**Clinical diagnosis**	
HFMD	215 (58.4%)
herpangina	10 (2.7%)
exanthema/rash/petechiae	127 (34.5%)
atypical HFMD	14 (38%)
**Side-symptoms**	
onychomadesis	16 (4.3%)
acute gastroenteritis	2 (0.5%)
respiratory symptoms	3 (0.8%)
NCS involvement (meningitis, encephalitis or paralysis)	7 (1.9%)

HFMD, hand-foot-mouth disease; SD, standard deviation.

**Table 2 viruses-13-00781-t002:** Comparison between different EV type-infections and clinical symptoms presented by the patients included in the study.

	EV Genotypes						
Clinical Diagnosis	CVA6 (N = 197)	CVA16 (N = 25)	CVA10 (N = 9)	EV-A71 (N = 32)	Other CVA (N = 12)	EV-B (N = 47)	*p* Values
HFMD (N = 215)	137 (63.7)	13 (6)	4 (1.9)	28 (13.0)	7 (3.2)	10 (4.6)	<0.00001
herpangina (N = 10)	0	2 (20)	3 (30)	1 (10)	1 (10)	1 (10)	0.5820
exanthema/rash/pethechiae (N = 127)	47 (37)	9 (7.1)	2 (1.6)	3 (23.6)	4 (3.1)	36 (28.3)	0.1812
atypical HFMD (N = 14)	13 (92.8)	1 (7.1)	0	0	0	0	0.00003
**Side-symptoms**							
onychomadesis (N = 16)	6 (37.5)	1 (6.2)	0	1 (6.2)	5 (31.2)	2 (12.5)	0.7289
NCS involvement (N = 7)	0	0	0	5 (71.4)	0	0	0.0209

HFMD, hand-foot-mouth disease; CV-, coxsackievirus; EV-, enterovirus. Percentage in brackets (%).

## Data Availability

Not applicable.

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
