# Peer review of "Surveillance for Enteroviruses Associated with Hand, Foot, and Mouth Disease, and Other Mucocutaneous Symptoms in Spain, 2006–2020"

_viruses, 2021, doi:10.3390/v13050781_

Round 1
Reviewer 1 Report
This manuscript "Surveillance for enteroviruses associated with hand, foot and mouth disease and other muco-cutaneous symptoms in Spain, 2006-2020" by Martinez-Lopez is nicely and very clearly written. It describes epidemiology of 368 EV-positive hospitalised patients with mucocutaneous symptoms identified in Spain over 15-year study period. Please find my minor comments below:
- Coxsackievirus A6 should be shortened as "CVA6" without the "-", whereas EV-A71 and EV-D68 are to be written with the "-" as in these cases the letter defers to species.
- It is important to note that authors have not investigated circulation of CVA6, but simply focused on its detection in clinical cases. Please modify line 69 and line 220 in discussion. This is important point as based on serological evidence CVA6 was likely circulating widely prior to 2008 but only noted based on clinical surveillance after that time point.
- Please explain how HFMD and atypical HFMD were defined in methods, and explain how competent you are that physicians have been adequately classifying these symptoms? Please modify accordingly in discussion line 272 onwards if required.
- In figure 1, authors state that they have seen CVA7... please confirm this finding - are you sure that these strains were CVA7 which used to be called as "poliovirus 4" based some older Russian data and I am not aware that this serotype would have been identified by any EV surveillance for years..
Author Response
This manuscript "Surveillance for enteroviruses associated with hand, foot and mouth disease and other muco-cutaneous symptoms in Spain, 2006-2020" by Martinez-Lopez is nicely and very clearly written. It describes epidemiology of 368 EV-positive hospitalised patients with mucocutaneous symptoms identified in Spain over 15-year study period. Please find my minor comments below:
- Coxsackievirus A6 should be shortened as "CVA6" without the "-", whereas EV-A71 and EV-D68 are to be written with the "-" as in these cases the letter defers to species.
The nomenclature has been corrected in the revised manuscripts as the reviewer suggested
- It is important to note that authors have not investigated circulation of CVA6, but simply focused on its detection in clinical cases. Please modify line 69 and line 220 in discussion. This is important point as based on serological evidence CVA6 was likely circulating widely prior to 2008 but only noted based on clinical surveillance after that time point.
The reviewer is right. The sentences have been changed in the new version of the manuscript (lines 72, 225 and 230).
- Please explain how HFMD and atypical HFMD were defined in methods, and explain how competent you are that physicians have been adequately classifying these symptoms? Please modify accordingly in discussion line 272 onwards if required.
HFMD and atypical HFMD have been defined in the revised manuscript as referee suggested (page 2, lines 86-92). We think that the pediatrician that perform the clinical diagnosis know the difference between typical and atypical HFMD but probably not all of them specified it when they requested the virological diagnosis. The sentence of Discussion has been changed (page 8, lines 291-293)
- In figure 1, authors state that they have seen CVA7... please confirm this finding - are you sure that these strains were CVA7 which used to be called as "poliovirus 4" based some older Russian data and I am not aware that this serotype would have been identified by any EV surveillance for years.
CVA7 from 2006 were characterized by neutralization assay using LBM antiserum pool A-H and they were not confirmed by sequencing in that moment. We have just repeated the typing using VP1 RTR-PCR and sequencing and the results are echovirus 11. They are changed in the revised manuscript.
Reviewer 2 Report
Title: Surveillance for enteroviruses associated with hand, foot and mouth disease and other muco-cutaneous symptoms in Spain, 2006-2020
The present manuscript provides surveillance study on the clinical samples of Hand, foot and mouth disease (HFMD). Authors analyzed the types, number of cases or cases depending on years, season or clinical signs. However, several revisions are required as below.
(Major revisions)
<Materials and methods>
- Line 79-103,
The contents are mixed of Results and M&M. You should remove the result and summarize the information of clinical samples. The proportion of patients or clinical diagnosis is suitable for Results section. It even contains the result of Fig 1 (line 85-89).
- Line 104-115,
Explain about the process of genotyping and sequencing in more detail. For example, pretreatment of clinical samples (if necessary), RNA extraction, nested RT-PCR (primers) for genotyping, cDNA synthesis and PCR (for sequencing), Sequencing…
- line 111-112
It is needed to show the summary of the sequences (strain and Genbank accession no.) used in phylogenetic analysis using a table.
- Why did you use 3’-VP1 region (400bp) not VP1 region in phylogenetic analysis? Normally, VP1 region is used in the genetic analysis of picornavirus.
<Results>
- Line 123,
What is the diagnostic method for EV-positive samples? RT-PCR? The information of EV-positive samples is needed in materials and methods.
- line 139~ (section 3.2.)
The result for Fig 1 should be move to Section 3.1. (line 142-148)
Moreover, the sentences for results and discussion are mixed in section 3.2. Separate and sort the sentences.
- line 150-156
The additional Figure or Table is need for the result. I cannot find the data on the age of patients in this manuscript.
- (Fig 1) Numbers of cases in 2020 is counted until March not 1 year (Discussion) in this study. It is better to edit 2020 to 2020 (Jan-Mar) in X axis.
- line 173-176
I think that the sentences is suitable for Discussion.
- (Table 1) 0.0000 means P<0.0001? It is better to write the numbers of P value exactly.
- line 181-193
The result of previous studies should be in discussion.
- High resolution file and remarkable colors and letters are needed in Fig 3 because readers cannot analyze.
<Discussion>
I think that the important discussion is missing.
Why is that the number of EV-pos cases are highest in 2016?
Why is that the number of EV-pos cases are highest in June and November, especially November?
What is the outbreak situation of E2 linage?
What is the reason that most Spanish strains are including in the E2 linage?
(Minor revision)
The abbreviation of Standard Deviation is SD?
Line 113, the reference of Clustal W is needed.
Author Response
Title: Surveillance for enteroviruses associated with hand, foot and mouth disease and other muco-cutaneous symptoms in Spain, 2006-2020
The present manuscript provides surveillance study on the clinical samples of Hand, foot and mouth disease (HFMD). Authors analyzed the types, number of cases or cases depending on years, season or clinical signs. However, several revisions are required as below.
(Major revisions)
<Materials and methods>
1. Line 79-103, The contents are mixed of Results and M&M. You should remove the result and summarize the information of clinical samples. The proportion of patients or clinical diagnosis is suitable for Results section. It even contains the result of Fig 1 (line 85-89).
Part of the section “patients and methods” has been moved to results in the revised manuscript, as the reviewer suggested (page 3, lines 121-122; page 5, Lines 156-159).
2. Line 104-115, Explain about the process of genotyping and sequencing in more detail. For example, pretreatment of clinical samples (if necessary), RNA extraction, nested RT-PCR (primers) for genotyping, cDNA synthesis and PCR (for sequencing), Sequencing…
We think that it not necessary to describe all the techniques in this article since they are published previously (references 30, 31).
3. line 111-112. It is needed to show the summary of the sequences (strain and Genbank accession no.) used in phylogenetic analysis using a table.
A new table (table 3) with strain and GB accession number has been added as supplementary material.
4. Why did you use 3’-VP1 region (400bp) not VP1 region in phylogenetic analysis? Normally, VP1 region is used in the genetic analysis of picornavirus.
For EV genotyping, partial or whole VP1 region can be used. Sequencing of the 3’-VP1 region (400 bp) amplified in our study is enough to obtain the EV genotype, as it was confirmed in a previously published article (Casas el at, Journal of Medical Virology 2001;65:138-148)
<Results>
5. Line 123, What is the diagnostic method for EV-positive samples? RT-PCR? The information of EV-positive samples is needed in materials and methods.
A sentence clarifying this point has been added to the new manuscript (page 2, line 93-94)
6. line 139~ (section 3.2.) The result for Fig 1 should be move to Section 3.1. (line 142-148). Moreover, the sentences for results and discussion are mixed in section 3.2. Separate and sort the sentences
Results for Fig 1 have been moved to section 3.1 in the revised manuscript (page 3, lines 132-133), and the results of section 3.2 have been clarified as the reviewer suggested.
7. line 150-156. The additional Figure or Table is need for the result. I cannot find the data on the age of patients in this manuscript.
A new table (Table 1) with the clinical features of the patients was added to the revised manuscript.
8. (Fig 1) Numbers of cases in 2020 is counted until March not 1 year (Discussion) in this study. It is better to edit 2020 to 2020 (Jan-Mar) in X axis.
The change suggested by the reviewer has been included in the revised manuscript
9. line 173-176. I think that the sentences is suitable for Discussion.
The sentences have been moved to Discussion
10. (Table 1) 0.0000 means P<0.0001? It is better to write the numbers of P value exactly.
The table has been changed as the reviewer suggested
11. line 181-193 The result of previous studies should be in discussion.
The sentence has been moved to Discussion (page 8, lines 113-114)
12. High resolution file and remarkable colors and letters are needed in Fig 3 because readers cannot analyze.
The tree figure has been changed to try to make it more resolution but they are many sequences and it is difficult to distinguish them all perfectly. Countries and years have been added to clarify.
<Discussion>
I think that the important discussion is missing.
13. Why is that the number of EV-pos cases are highest in 2016?
Probably it was due to that year an outbreak of severe encephalitis cases caused by EV-A71 occurred in Spain. This fact alerted clinicians and microbiologists to EV infections and more EV tests were performed; subsequently, more EV-samples were sent to our lab for genotyping. This point has been added to the Discussion in the revised manuscript (page 7, lines 264-267).
14. Why is that the number of EV-pos cases are highest in June and November, especially November?
Seasonal circulation (spring and autumn) in temperate countries is widely documented. The highest circulation of HFMD-associated EV in June and November matches up overall EV circulation in our country, since the months of May-June and October-November are the ones that the lab receives the largest number of EV-positive samples. The explanation could be that In May-June it's already hot but there are still school classes; in October, classes have just begun. This point has been clarified in the revised manuscript (page 7, lines 269-274).
15. What is the outbreak situation of E2 linage? What is the reason that most Spanish strains are including in the E2 linage?
Spanish CVA6 sequences from 2010-2020 are genetically similar that those isolated in other countries during the same years and all of them fall into one or several lineages/clusters (depending on the authors) but they are related to each other so far. The underlying reasons of the emergence of this new lineage (or lineages) of CVA6 in 2008/2010 that has spread worldwide associated with a greater number of HFMD cases and/or different clinical presentations are unknown. These points have been discussed in the revised manuscript as the reviewer suggested (page 8, lines 311-322)
(Minor revision)
-The abbreviation of Standard Deviation is SD?
Yes, it has been corrected
-Line 113, the reference of Clustal W is needed.
The reference of Clustal W has been added (ref 32) in the revised manuscript
Reviewer 3 Report
The submission is a presentation of conventional description of cases as a case report without serious consideration for the interpretation of the findings with supportive previous work. Following are my general suggestions and comments:
- The presentation of dendrogram should be linked to the relevant factors of the host and their environment. Otherwise it is mainly a genetic presentation of the isolated viruses.
- Case definition including symptoms with the list of differential diagnosis were not presented in detail so that the readers can recognize the generalization and limitation of these recruited cases.
- Although the study is strictly descriptive in its type, statistical analysis of the data is required to determine the association of specific case characteristics to the type of isolated viruses; otherwise no inference can be made.
- I wish the authors includes some matching non-cases so that potential risk factors can on the recruitment of the cases with the limitations of those cases to represent the population under this study. Making inference from their findings is limited since no comparison was presented.
Author Response
The submission is a presentation of conventional description of cases as a case report without serious consideration for the interpretation of the findings with supportive previous work. Following are my general suggestions and comments:
1. The presentation of dendrogram should be linked to the relevant factors of the host and their environment. Otherwise it is mainly a genetic presentation of the isolated viruses.
Unfortunately, information about host and environmental factors is not available. The phylogenetic analysis only provides molecular epidemiology of the virus
2. Case definition including symptoms with the list of differential diagnosis were not presented in detail so that the readers can recognize the generalization and limitation of these recruited cases.
Definitions of the different muco-cutaneous presentations have been added to the revised manuscript (page 2, lines 86-92)
3. Although the study is strictly descriptive in its type, statistical analysis of the data is required to determine the association of specific case characteristics to the type of isolated viruses; otherwise no inference can be made.
This study is a retrospective and descriptive study about different serotypes identified in EV infections causing muco-cutaneous diseases. Statistical analysis (comparison of proportions) was performed using Chi-square test and P values are indicated in Table 2.
4. I wish the authors includes some matching non-cases so that potential risk factors can on the recruitment of the cases with the limitations of those cases to represent the population under this study. Making inference from their findings is limited since no comparison was presented.
This study is a surveillance study about which EV types are implicated in muco-cutaneous infections in Spain. As it is mentioned in Section of methods, for NPEV surveillance, our lab received EV-positive samples from different Spanish hospitals voluntarily and at the discretion of clinicians. Data about real incidence of these infections are no available and the results could be little biased but our lab has been performing EV surveillance for over 30 years and with a mean of 500 samples analyzed each year, we are convinced that the data on which EV circulates and what pathologies produce in our country are quite close to reality.
Round 2
Reviewer 2 Report
Thank your for your efforts and I suggest minor revisions.
- (Fig 1) The numbers of X axis (2006, 2007, 2008...., March 2020) are missing.
- (Fig 3) The legend or description of colors on years are needed because readers cannot easil recognize years(strain names) in phylogenetic trees.
Author Response
1. (Fig 1) The numbers of X axis (2006, 2007, 2008...., March 2020) are missing.
The figure 1 has been corrected.
2. (Fig 3) The legend or description of colors on years are needed because readers cannot easil recognize years(strain names) in phylogenetic trees.
The colors of the years of the sequences have been added to the Figure 3 legend.
Reviewer 3 Report
The authors have improved the presentation of this submission and they considered all suggestions and comments from the previous review, Good luck in finalizing the submission.
Author Response
The authors have improved the presentation of this submission and they considered all suggestions and comments from the previous review, Good luck in finalizing the submission.
Thank you very much